# Beyond Calories: Addressing Micronutrient Deficiencies in the World’s Most Vulnerable Communities—A Review

**DOI:** 10.3390/nu17243960

**Published:** 2025-12-18

**Authors:** James Ayokunle Elegbeleye, Olanrewaju E. Fayemi, Wisdom Selorm Kofi Agbemavor, Srinivasan Krishnamoorthy, Olalekan J. Adebowale, Adeyemi Ayotunde Adeyanju, Busisiwe Mkhabela, Oluwaseun Peter Bamidele

**Affiliations:** 1Department of Microbiology, Faculty of Sciences, Southwestern University, Nigeria, Kilometre 20, Sagamu-Benin Expressway, Okun-Owa 120104, Ogun State, Nigeria; 2Department of Public Health, College of Allied Health Sciences, Mountain Top University, Kilometre 12, Lagos-Ibadan Expressway, Prayer City 110106, Ogun State, Nigeria; 3Radiation Technology Centre, Biotechnology and Nuclear Agriculture Research Institute, Ghana Atomic Energy Commission, Legon, Accra P.O. Box LG 80, Ghana; 4National Institute of Food Technology, Entrepreneurship and Management, Thanjavur (NIFTEM-T), Thanjavur 613005, Tamil Nadu, India; 5Department of Food Technology, The Federal Polytechnic, Ilaro 112106, Ogun State, Nigeria; 6Centre for Innovative Food Research (CIFR), Department of Biotechnology and Food Technology, Faculty of Science, University of Johannesburg, Johannesburg 2092, Gauteng, South Africa; 7Department of Food Science and Microbiology, College of Pure and Applied Sciences, Landmark University, Omu-Aran 251103, Kwara State, Nigeria; 8Department of Nutrition, University of Venda, Thohoyandou 0950, Limpopo, South Africa; 9Department of Food Science and Technology, University of Venda, Thohoyandou 0950, Limpopo, South Africa

**Keywords:** malnutrition, micronutrient deficiencies, hidden hunger, vulnerable populations, food fortification

## Abstract

Micronutrient deficiencies, also known as “hidden hunger,” remain a pervasive public health issue in low- and middle-income countries, particularly among vulnerable populations within these countries. The main drivers of these deficiencies are poverty, limited dietary diversity, weak nutritional strategies, poor health service delivery and general health access barriers. This review assesses the prevalence, drivers, and consequences of selected micronutrient deficiencies: iron, iodine, zinc, vitamin A and vitamin D, within the scope of undernutrition, food insecurity, and socioeconomic inequity. The consequences associated with these deficiencies include stunted growth, increased susceptibility to illness, poor cognitive and social functioning, and deepened poverty. The primary strategies to address these deficiencies include dietary diversification, supplement provision, biofortification, and the production of fortified foods. Barriers to progress include the high cost of food, weak healthcare infrastructure, low educational levels, and ineffective policy implementation. Integrated food systems, personalised nutrition, and innovative food technologies have the potential to address both nutritional and health inequities. Addressing barriers to safe and nutritious food and healthcare systems in order to address health inequities requires integrated, multisectoral planning and contextual policy. Improving individual health outcomes is crucial, but addressing micronutrient deficiencies has a ripple effect throughout society, enabling economic development through poverty reduction and increased productivity.

## 1. Introduction

Micronutrient deficiencies, often referred to as “hidden hunger,” are a significant health crisis that affects an estimated 2 billion people globally, with the majority of affected populations residing in low- and middle-income countries (LMICs) [1]. While global attempts have successfully reduced protein-energy malnutrition and its associated effects, micronutrient deficiency persists, resulting in severe physical, cognitive, and socioeconomic impacts. Hidden hunger is a significant aspect that has the potential to undermine the achievement of several Sustainable Development Goals (SDGs), particularly those related to health, education, and poverty eradication [2].

The continued burden of micronutrient deficiencies in LMICs is strongly linked to limited dietary diversity driven by economic hardship, reduced food accessibility, and technological gaps in food systems [3]. Overcoming these challenges requires a multi-pronged approach. Micronutrient deficiencies occur when an individual’s diet lacks one or more essential vitamins or nutrients [4], posing serious health risks for vulnerable populations, including children, pregnant or lactating women, the elderly, refugees, and low-income households. Young children (0–5 years) are particularly prone to stunting, wasting, and cognitive impairment, especially among malnourished infants and low-birth-weight babies [5]. Pregnant and lactating women face increased risks of anaemia, maternal mortality, and birth complications, particularly in food-insecure households [6]. Among older adults, poor dietary intake weakens immunity and increases susceptibility to chronic diseases, particularly among individuals living in poverty or experiencing social isolation [7,8]. Refugees and displaced persons frequently endure severe food insecurity, malnutrition, and disease outbreaks due to conflict-related instability [9]. Low-income families often rely on nutrient-poor diets due to limited financial resources or residence in food deserts, which further exacerbates hidden hunger [10]. Even mild malnutrition in children impairs concentration, learning, and long-term productivity [11]. Poor nutrition, more broadly, diminishes physical and cognitive abilities, thereby perpetuating cycles of poverty [12]. Table 1 summarises major micronutrient deficiencies, their causes, and the populations most affected.

Globally, hunger remains a significant issue, even more so for socially vulnerable groups like children, pregnant and lactating women, the elderly, refugees, and people living in poverty [10]. The different groups are vital intervention targets in improving the public health of a population, social cohesion, and public safety. Hunger, along with malnutrition, has a devastating toll on public health, worsening the malady in chronically vulnerable groups like children, women during and after pregnancy, and elderly persons. Chronic hunger is directly responsible for stunting a child’s growth, weakening immunity, and cognitively impairing them, hindering their prospects and life opportunities [13].

This review critically examines the global burden and strategic interventions for several critical micronutrient deficiencies, including iron, Zinc, Calcium, Magnesium, and vitamins A, D, and E. They were selected based on their significant health burdens, high prevalence, and the availability of scalable intervention models. We evaluate immediate clinical strategies in conjunction with broader, long-term public health approaches, including food fortification, biofortification, supplementation, and policy measures. By reviewing the effectiveness of these interventions, we propose evidence-based perspectives to support stronger and more sustainable global efforts to combat micronutrient deficiencies.

**Table 1 nutrients-17-03960-t001:** Micronutrient deficiencies and their causes.

Micronutrient	Deficiency Disease/Impact	Causes	References
Iron	Anaemia, fatigue, impaired cognitive function	Inadequate dietary intake, blood loss (menstruation, ulcers), poor absorption (intestinal disorders)	[14]
Vitamins	Night blindness, weakened immunity, and childhood mortality Rickets, osteoporosis, and muscle weakness	Insufficient intake of animal and plant-based sources, malabsorption disorders, and protein deficiency. Limited sun exposure, poor dietary intake, and kidney/liver disorders affect vitamin D metabolism. Insufficient fruit and vegetable intake, smoking (increases vitamin C requirement). Malabsorption disorders, long-term antibiotic use, and liver disease	[15,16]
Iodine	Goitre, cognitive impairment, hypothyroidism	Low iodine in soil and water, lack of iodised salt consumption	[17]
Zinc	Growth retardation, impaired wound healing, weakened immunity	Not consuming enough zinc-rich foods like meat, seafood, and poultry. Consuming foods high in phytates	[18]
Calcium	Osteoporosis, muscle cramps, weak bones	Low dairy intake, vitamin D deficiency (affecting calcium absorption), high sodium/caffeine intake (increases calcium loss)	[19]

## 2. Global Burden of Micronutrient Deficiencies: Prevalence, Drivers, and Consequences

Micronutrient deficiency encompasses deficiencies in essential vitamins and minerals, typically resulting in insidious and widespread outcomes. This differs from macronutrient deficiency, which primarily involves insufficient intakes of nutrients such as protein, carbohydrates, as a person suffering from hidden hunger may not exhibit any signs of starvation. However, it can lead to severe health impacts, as well as economic and social effects that transcend the individual [20].

### 2.1. Prevalence of Micronutrient Deficiencies and Their Impact on Health Globally

Micronutrient deficiencies are a public health concern that affects almost every country worldwide, especially LMICs [21,22]. Although high-income countries are less affected, they are not completely exempt from the impacts. The most common micronutrient deficiencies in people are iron, vitamin A and D, iodine, and zinc [23]. According to the World Health Organisation (WHO), more than two billion people globally suffer from one or more micronutrient problems [21]. The various elements of hidden hunger are further elaborated below.

#### 2.1.1. Iron Deficiency

Iron deficiency is the leading form of micronutrient deficiency and is prevalent among women of reproductive age, pregnant women, and young children [24]. Failing to meet the dietary requirements of iron, which are crucial for producing haemoglobin and transporting oxygen, leads to its deficiency. Some reasons that could lead to Iron deficiency include inadequate intake from the diet, blood loss through menstruation, parasitic infection from hookworms, malabsorption due to celiac disease, and increased physiological requirements during growth and pregnancy [25], In some cases, minerals are present in the diet but remain inaccessible to the body because they are bound to other compounds like phytic acid, an antinutrient, within the food matrix. This binding prevents their absorption, contributing significantly to deficiencies of essential minerals such as iron and zinc. This issue is particularly prevalent in areas where diets are predominantly based on cereals and pulses [22]. The primary consequence of this micronutrient deficiency (Iron) is anaemia, as well as fatigue, weakness, and cognitive impairment. Studies showed children who are anaemic tend to lose 5–10 IQ points and lower growth potential, whereas for pregnant women, they become more susceptible to preterm delivery and increased risk of maternal mortality during pregnancy [26]. Chronic iron deficiency can significantly hinder work productivity and weaken the immune system. This deficiency predominantly affects women and children from low- and middle-income countries, especially from sub-Saharan Africa and South Asia, where diets lack iron-rich foods and parasitic infections are rampant [27]. The challenges of iron deficiency must be addressed, including compliance with supplemental iron and equitable access to fortified foods, as well as diversified diets [28].

#### 2.1.2. Vitamin A Deficiency

Low- and middle-income countries, especially those in sub-Saharan Africa and South East Asia, are home to 190 million preschool-aged children and 19 million pregnant women suffering from Vitamin A Deficiency (VAD) [29]. It is one of the most common nutritional deficiencies. It occurs when insufficient dietary consumption of vitamin A, retinol, or its precursor, beta-carotene, restricts one’s eyesight, immune system, and growth [30]. Limited access to vitamin A sources like liver, dairy, orange-coloured vegetables, low dietary variety, impaired absorption caused by diarrhoea or parasitic infections, and increased requirements due to growth or pregnancy cause Vitamin A deficiency [30].

VAD is the leading cause of preventable blindness in children, resulting in night blindness and causing xerophthalmia [31]. It also impairs immune responses, increasing morbidity and mortality from infections such as measles and diarrhoea. For pregnant women, it increases the risk of maternal mortality and poor outcomes. In high-risk regions, it affects roughly 30% of children under five years of age, fuelled by poverty, food insecurity, and dependence on staple crops low in Vitamin A [32].

#### 2.1.3. Vitamin D Deficiency

There are currently as many as 1 billion people affected by Vitamin D deficiency, which is defined as a global health concern [33]. It is alarming, however, that both high and low-income countries suffer from this problem. Deficiency occurs due to limited exposure to sunlight and inadequate dietary intake, resulting in the impaired body’s ability to absorb crucial Vitamin D [34]. Insufficient exposure to sunlight is especially prevalent in high-latitude regions, where individuals often spend most of their time indoors and wear full-body clothing [35]. Moreover, the lack of vitamin D-rich foods such as fortified dairy and fatty fish compounds the issue. Deficiency increases nutrient malabsorption (e.g., impairs Vitamin D absorption) in aged individuals and those going through pregnancies, as well as in those with conditions such as celiac disease and Crohn’s disease [36]. Darker skin pigmentation also significantly impairs vitamin D synthesis [35].

A deficiency of vitamin D in children can cause rickets, resulting in skeletal deformities. It causes osteomalacia in adults and increases the risk of osteoporosis [37]. A lack of vitamin D further increases the risk of developing cardiovascular disease, diabetes, certain forms of cancer, and autoimmune disorders [38]. Additionally, it reduces the immune response, which increases the risk of infection, including respiratory infections. Targets include nursing infants, elderly individuals, and those who spend little time outdoors [39].

#### 2.1.4. Deficiency of Iodine

Iodine deficiency remains one of the most common and widespread global issues, affecting approximately 1.9 billion people worldwide, with a disproportionately larger impact on low- and middle-income countries [40]. This deficiency occurs due to inadequate dietary consumption of iodine, which is essential for the thyroid gland’s requirements, fundamental for metabolism, growth, and neurologic development [17]. Goitrogenic foods (such as cassava), low salt intake, and limited access to dairy products and seafood are major contributors to iodine deficiency [41]. Iodine deficiency has several consequences, such as impaired thyroid function, goitre (enlarged thyroid) and hypothyroidism. In females, this could cause miscarriage and congenital hypothyroidism from either iodine or thyroid deficiency [42]. Severe iodine deficiency in pregnancy causes maternal cretinism, characterised by intellectual disability and dwarfism due to irreversible brain damage [43] and severe impaired cognitive development (decline in IQ by 10–15 points). Even Mild iodine deficiency can hinder learning and productivity [44].

#### 2.1.5. Zinc Deficiency

Globally, roughly 17% of the population is affected by zinc deficiency, and the figure is comparatively significant for low- and middle-income countries, especially for countries located in Sub-Saharan Africa and South Asia [45]. Zinc deficiency occurs when there is insufficient zinc consumed and absorbed in the diet to fuel necessary physiological processes such as immune function, growth, and DNA replication [46]. An increase in the consumption of phytate-laden staple foods such as cereals and legumes, coupled with limited uptake of zinc-laden foods like meat and shellfish, plus increased consumption during growth, pregnancy, or illness, also contributes to higher zinc deficiency rates [47].

A deficiency of Zinc weakens the immune response, which increases the risk of infections such as pneumonia and diarrhoea, major contributors to child mortality [48]. Zinc deficiency is associated with slow growth, increased time to reach reproductive age, various skin diseases, and stunted growth. For women, the consequence is an escalated risk of premature labour and decreased weight of the child during delivery. Zinc deficiency is frequent in areas where various foods are not consumed. In these areas, it impacts 30% of expecting mothers and mothers [49]. Table 2 shows the estimated global prevalence of key micronutrient deficiencies by population group.

**Table 2 nutrients-17-03960-t002:** Global Prevalence of Key Micronutrient Deficiencies by Population Group (2010–2022 Estimates).

Micronutrient Deficiency	Population Group/Diagnostic Criteria	Global Prevalence Estimate	Notes/Year of Data	Primary Reference(s)
Iron (Anaemia)	Children 6–59 months	40% (269 million children)	2019 data; Anaemia is often used as a proxy for the iron deficiency burden.	[50]
Iron (Anaemia)	Pregnant Women (15–49 years)	37% (32 million women)	2019 data. Regions of Africa and South East Asia are most affected.	[50]
Iron (Anaemia)	Non-pregnant Women (15–49 years)	30% (539 million women)	2019 data. Dietary iron deficiency causes 66.2% of total anaemia cases (2021).	[50]
Iodine Deficiency Disorders (IDD)	Women of Reproductive Age	81.4 million Prevalent Cases	2019 GBD estimate. The age-standardised prevalence declined by 13.3% since 1990.	[51]
Iodine (Access)	Global Households	89% using iodised salt	The 2020 estimate indicates nearly 1 billion people lack consistent access.	
Vitamin A Deficiency (VAD)	Preschool-aged Children (0–5 years)	19.53% (VAD, low serum retinol)	Systematic review finding (2023). Classified as a mild/moderate PH problem globally.	[52]
Vitamin A Deficiency (Severe PH Problem)	Preschool-aged Children (Africa Region)	30.59%	The highest regional prevalence was observed in this age group, indicating a severe public health crisis.	[52]
Zinc (Risk of Inadequate Intake)	Global Population (All ages)	17.3%	Based on food balance sheets and theoretical physiological requirements (IZiNCG).	[53]
Vitamin D (Deficiency)	Global Population (All ages)	15.7%	Serum 25(OH)D < 30 nmol/L. Data pooled from 2000–2022 systematic review.	[54]
Vitamin D (Insufficiency)	Global Population (All ages)	47.9%	Serum 25(OH)D < 50 nmol/L. Nearly half the global population is affected.	[54]
Folate Deficiency	Women of Reproductive Age (Lower-Income Economies)	>20%	Systematic review finding. Highlights stark economic disparity compared to HICs (<5%).	[55]
Polynutritional Deficiency	Children under five years old	>50%	An estimate of deficiency in at least one key nutrient (Iron, Zinc, or Vitamin A).	[56]

### 2.2. The Challenge of Diagnosis and Screening in Vulnerable Settings

The problem of hidden hunger, as a global health burden, may be underestimated due to significant diagnostic challenges, particularly in vulnerable communities where these deficiencies are most prevalent. This assumption is more apparent in the non-specific nature of the symptoms associated with micronutrient deficiencies, unlike those associated with macronutrient deficiencies [57]. Important biomarkers used as the gold standard, like serum 25-hydroxyvitamin D [25(OH)D] for vitamin D, serum ferritin for Iron, or plasma zinc in the laboratory assessment of these deficiencies, require specific equipment, expertise and stable low temperature of the samples, which are absent in most LMICs [58].

In a resource-limited setting, where clinical signs and proxy measures (e.g., anaemia for Iron deficiency) are relied upon, there is a high risk of inaccuracy in assessment, misdiagnosis, and delayed intervention [59]. A significant cost-effectiveness gap also exists in the development and deployment of an affordable, non-invasive, point-of-care screening approach used in targeted interventions and surveillance by public health workers.

## 3. Vulnerable Populations at Risk of Malnutrition

### 3.1. Children and Adolescents

Micronutrient deficiency-based disorders are expected to affect between 1.6 and 2 billion people worldwide [60]. Women, including adolescents and children, bear the majority of the burden [61]. Notably, a compromised immune system causes at least 157,000 per annum early childhood diseases such as diarrhoea, measles, malaria, and other infections. Meanwhile, inadequate vitamin A intake causes about 350,000 cases of childhood blindness, with half of those affected passing away within a year of losing their sight [62]. According to estimates, deficiencies in zinc and vitamin A cause the deaths of 1.1 million children under five per year. In 2016, it was projected that more than 300,000 children were born with severe birth abnormalities annually because of maternal folate insufficiency [63]. The much higher rates of maternal death, stillbirth, and neonatal mortality are further consequences of the variable diets and reduced health and well-being of women and young children. From the economic point of view, an estimated 2% to 5% of GDP is lost annually due to micronutrient deficiencies alone [64].

### 3.2. Pregnant and Lactating Women

The World Health Assembly’s (WHA) 2025 estimated 9.7% underweight and 32.8% anaemia among women of reproductive age (15–49 years) worldwide, which were 5% and 15%, respectively, in 2016 [65]. In addition to contributing to more than 600,000 stillbirths or neonatal deaths and more than 100,000 maternal deaths during pregnancy, malnutrition puts 40% to 60% of children in the 6–24-month age group at risk for less-than-optimal development [61]. About 20% of childhood stunting is reported to be caused by maternal malnutrition, which also puts mothers at risk for maternal difficulties and exposes infants to low birth weight, foetal birth abnormalities, limited physical and mental development, and foetal or newborn mortality [62]. An undernourished mother has a higher chance of giving birth to a child who will also be undernourished as an adult. Therefore, preventing malnutrition among nursing mothers is crucial to breaking the cycle of malnutrition in the population. Meanwhile, it is estimated that maternal iodine insufficiency causes over 18 million infants to be born with intellectual disabilities [63]. The major tasks ahead are addressing these challenges through approaches such as fortification and functional food applications.

The costs of micronutrient malnutrition vary by the socioeconomic status of the subpopulation within a country. For instance, in the Philippines, the costs of micronutrient deficiencies in the poorest third of households were estimated to be five times higher than in the wealthiest third. These discrepancies exacerbate the financial strain on health systems, which are often overburdened and underfunded [66].

### 3.3. Elderly

Malnutrition in older adults has been recognised as a challenging health issue affecting a wider population worldwide. It has been linked to physical deterioration that impacts daily activities and overall quality of life, as well as increased mortality and morbidity. Clinical guidelines for assessing and managing malnutrition in the older population are provided by major organisations worldwide, such as the European Society for Clinical Nutrition and Metabolism (ESPEN) [67]. These recommendations cover the fundamentals of regular screening, precise evaluation, and tailored intervention strategies. Regular screening for malnutrition using instruments such as the Mini Nutritional Assessment (MNA) and the Malnutrition Universal Screening Tool (MUST) is recommended by the 2023 ESPEN standards, which are adhered to in European nations [68]. These instruments help determine who is more likely to suffer from malnutrition and enable prompt treatment to lessen the possible suffering brought on by this illness.

Unintentional weight loss or a low body mass index, particularly in cases of chronic diseases and functional impairments, has been an indicator of this serious public health issue impacting older people. Immune function impairment has been strongly associated with malnutrition-related protein catabolism and cell-mediated immunity, especially in older malnourished people, which raises the risk of infection and slows any healing process [69]. Consequently, several studies have demonstrated a strong correlation between malnutrition and infection risk, including infections linked to healthcare, infectious complications, and the ensuing lengthier hospitalisations in intensive care units (ICU), and higher ICU mortality in older malnourished patients [62,70,71]. Malnutrition, which is partially triggered by a lack of certain micronutrients, also impairs wound healing and tissue recovery. This clearly puts older people at risk for chronic wounds and wound healing problems, which are very burdensome for older adults and are linked to lower quality of life and increased health care costs.

Sadly, the existing diagnostic criteria for identifying malnutrition in older individuals are still insufficient and frequently overlooked at the onset of the condition. Even while weight loss, BMI, and albumin levels are currently utilised for diagnosis, they can be impacted by things like inflammation or dehydration, which can result in inaccurate diagnoses.

### 3.4. Low-Income Communities

Similarly, low-income families are more susceptible to malnutrition because they have less access to wholesome food and quality medical treatment, which frequently leads to undernutrition. Poverty eradication is the first of all the 17 Sustainable Development Goals in all its manifestations. The second target is to eradicate hunger, reduce food insecurity, and enhance agricultural productivity and nutrition. Additionally, indications that are extremely pertinent to nutrition are included in at least 12 of the 17 goals [72]. It is crucial to address poverty and hunger at the same time rather than in separate silos since they are closely linked and each feeds the other. A vicious cycle of starvation, health issues, and decreased productivity is created by poverty, a lack of resources, and poor sanitation.

Particularly, the youngsters have a detrimental effect on their physical, mental, and emotional development. Poverty has a variety of effects on people, such as inadequate nutrition, food insecurity, increased susceptibility to illness, decreased productivity, and stunted intellectual and physical growth. Furthermore, those in poverty lack access to basic needs like wholesome food, a clean environment, suitable housing, and quality medical care. Given the aforementioned connections between poverty and malnutrition, nutrition-specific interventions must be supplemented with nutrition-sensitive approaches to accelerate the reduction of malnutrition. Interventions that target the intermediate and underlying causes of malnutrition and contribute to better access to wholesome food, clean water, sanitation, work, education, healthcare, etc., are known as nutrition-sensitive interventions. To increase the coverage and efficacy of nutrition-specific interventions, large-scale nutrition programs that emphasise evidence-based nutrition interventions should simultaneously focus on important underlying determinants of nutrition, such as poverty [73].

### 3.5. Refugees and Displaced Persons

Meanwhile, one of the vulnerable populations affected by micro and macro nutrient disorders is refugees and displaced persons. The number of people who have been forcibly displaced by war, violence, conflict, or persecution is at least 26 million worldwide. In host nations worldwide, the great majority of refugees (about 78%) reside in cities and metropolitan areas as opposed to refugee camps [74]. The majority of refugees lived in nations or regions with severe food shortages and malnutrition before relocation. Low- and middle-income nations are particularly vulnerable to nutritional issues, including the double burden of malnutrition, and host a significant number of refugees. Particularly, malnutrition is a common issue among refugees because of their shift to radically different food systems. Displaced people frequently face barriers to eating healthy diets in the host nations due to economic, cultural, and language barriers [75].

Mass displacement is typically linked to changes in the prevalence of malnutrition among migrants during transfer from one nation to another. Such changes in nutritional status have significant medical, developmental, economic, and social implications for both the countries and the migrants and their families [76]. Environmental and socioeconomic factors, such as low income, unemployment, insufficient access to clean water and healthcare, limited food and humanitarian resources, may also exacerbate nutrition deficiencies among migrants and refugees in industrialised nations [9]. Perhaps the refugees, while adjusting to their host nations, experience changes in their nutritional state during the pre- and post-resettlement stages. For refugees to improve their diet and nutrition, a complete dietary and health evaluation is required, along with culturally relevant and long-lasting nutrition education materials and interventions. To measure changes in the food intake and nutritional status of refugees and to further explore aspects related to these two components, longitudinal studies and innovative methodological approaches are also recommended.

## 4. Relationship Between Micronutrient Deficiencies and Hunger

### 4.1. Hunger as a Contributor to Nutrient Deficiencies

Hunger, particularly food insecurity, often results when a diet is deficient in vital nutrients and required energy. Notably, the energy source needed to power every bodily function is nutrition. A combination of macro- and micronutrients makes up a balanced diet. Noteworthily, “nutritional deficiency” refers to significantly low levels of one or more nutrients that prevent the body from functioning normally, and “nutritional inadequacy” refers to a nutrient intake that is below the estimated average requirement [76]. Micronutrient deficiencies and food poverty have a complicated link because they necessitate the coordination of policies, programs, and initiatives that affect food systems and support dynamic processes from production to consumption. The worldwide syndemic of obesity, malnutrition, and even climate change, which is described as “the synergisms between pandemics that coexist in time and space, interact with each other, and share common core social factors”, is connected to the cyclical impact of food insecurity [74].

### 4.2. Socioeconomic Factors Influencing Hunger and Nutrition

Generally, diseases and/or insufficient food consumption, either in terms of quantity or quality, are the direct causes of malnutrition. However, several underlying poverty-related problems, such as inadequate water, sanitation, and health facilities, as well as food insecurity, have an impact on malnutrition [77]. Other socioeconomic factors include conflict, climate change, a lack of natural resources, unstable food prices, inadequate governance, and population expansion. Malnutrition also prolongs poverty and hampers economic growth. Malnutrition-related mortality and morbidity directly reduce the economy’s human capital and productivity. At the microeconomic level, it is estimated that a 1 per cent reduction in adult height due to childhood stunting translates into a 1.4 per cent decrease in an individual’s productivity [78].

Early childhood undernutrition leads to impaired cognitive function and lower school attainment, which result in additional indirect costs for the nation’s economy. In reality, the lack of education and the resulting lower worker skill levels significantly impede the development of nations affected by malnutrition. Malnutrition leads to an overall economic cost that varies from 2 to 3 per cent of GDP up to 16 per cent in the majority of impacted nations [79]. Malnutrition has long-lasting impacts that keep people and communities stuck in a vicious cycle of poverty for generations. Therefore, enhancing nutrition is crucial to ending poverty and boosting the economies of low- and middle-income nations.

### 4.3. Case Studies of Affected Populations

Eliminating micronutrient deficits is not best achieved by concentrating on a specific micronutrient deficiency or method. A broader range of elements, such as housing, water supply, sanitation, education, and health care, frequently contribute to the issues [79]. Comprehensive public health interventions, such as deworming, malaria prevention, improved water and sanitation facilities, and child vaccination, can assist in reducing micronutrient malnutrition. Strategies that tackle all of these problems in a coordinated and integrated manner are successful. The most effective methods for lowering micronutrient deficiency are holistic approaches that combine public health initiatives, education and awareness campaigns, and direct and indirect interventions [80]. A few case studies on the burden of affected people with hidden hunger and malnutrition challenges are detailed here.

India, with around one-third of the world’s population, faces a major challenge, suffering from micronutrient deficiencies. The nation faces the heartbreaking fact that over 6000 children under five die every day, and that malnutrition, more particularly, a lack of zinc, iron, vitamin A, and folic acid, is responsible for almost half of these deaths [81]. Moreover, in a recent survey study from 84 countries, the highest incidence of combined stunting and wasting (4·4%) due to undernutrition was found in South Asia, with an overall prevalence of 3·0% in surveys conducted between 2005 and 2015. The frequency of both stunting and wasting in children aged 6 to 59 months was greater than 5% in nine different nations. According to cohort analyses, South Asia had a much higher peak prevalence of concurrent wasting and stunting at age 2 (8%) than either Africa (2%) or Latin America (1%). A summary of micronutrient deficiencies, mainly vitamins, their consequences and the affected vulnerable population is shown in Table 3.

**Table 3 nutrients-17-03960-t003:** Summary of common micronutrient (mainly vitamins) deficiencies and vulnerable populations.

S/No	Micronutrient	Deficiency Explanation	Consequences	Vulnerable Population	References
1.	Vitamin A	Concentration of serum retinol or retinol binding protein below the cutoff value	Impaired vision, including night blindness and blindness in severe cases	Young children, particularly those under 5 years old, and pregnant women	[70]
2.	Thiamine	Low level of thiamine in blood and low activity of the enzyme “transketolase”	Several disorders, including neurological and cardiovascular issues	Infants, particularly those who are exclusively breastfed by mothers with thiamine deficiency, adults majorly pregnant and lactating women	[82]
3.	Riboflavin	Low level of riboflavin in the blood	Sore throat, lesions in the mouth, and skin disorders	Athletes, pregnant and lactating women, older persons, habitual alcohol abusers, and adults with liver disease	[83]
4.	Niacin	Urinary excretion of specific niacin metabolites, such as N1-methylnicotinamide (NMN).	Dermatitis, diarrhoea, and dementia	People experiencing poverty or malnutrition, those with anorexia nervosa, alcohol use disorder, AIDS, inflammatory bowel disease, or liver cirrhosis	[84]
5.	Folate	Low level of folic acid in the blood	Megaloblastic anaemia, fatigue, irritability, and neurological problems	Pregnant women, children under 5 years of age, and the elderly	[85]
6.	Cobalamin	Low level of cobalamin in blood	Impacting neurological function and the synthesis of red blood cells	Older adults, individuals with gastrointestinal disorders, infants of vegan mothers and women of reproductive age	[86]
7.	Calciferol	Lower levels of 25-hydroxyvitamin D in the blood	Soft and deformed bones, leg deformities, osteomalacia	older adults, people with darker skin, and infants	[87]
8.	Ascorbic acid	Lower levels of ascorbic acid in the blood	Scurvy, characterized by fatigue, weakness, bleeding gums, loose teeth, and skin changes like bruising and spots	Smokers, elderly people, children, pregnant women, and anyone with malabsorption problems	[88]
9.	Tocopherol	Lower levels of alpha-tocopherol in the blood serum	Muscle weakness, coordination problems (ataxia), peripheral neuropathy, and haemolytic anaemia	Premature infants, particularly those with very low birth weights, and individuals with impaired fat absorption, such as cystic fibrosis or abetalipoproteinemia	[89]

## 5. Challenges in Addressing Micronutrient Deficiencies

Despite global efforts to combat malnutrition, micronutrient deficiencies remain a significant public health concern, particularly among vulnerable populations. Addressing these challenges requires more than just increasing food availability; it involves tackling deep-rooted challenges related to accessibility, healthcare, education, policy formulation, and implementation. The following section explores key obstacles that impede progress in eradicating micronutrient deficiencies.

### 5.1. Awareness and Education

One of the basic challenges in tackling micronutrient deficiencies is the lack of awareness and education regarding proper nutrition. Most affected individuals, particularly in LMICs, have limited knowledge about the importance of micronutrients and how to incorporate them into their diets. Several factors contribute to this gap: Limited nutrition education remains a key barrier to improving micronutrient intake in many developing countries, where nutrition is often not included in school curricula. As a result, children and adolescents grow up unaware of the essential role that vitamins and minerals play in growth and development [90]. Cultural and dietary practices also significantly contribute to micronutrient deficiencies, especially in populations that rely heavily on carbohydrate-rich staple foods and have limited intake of protein and vegetables, thereby reducing dietary diversity [91]. In addition, misinformation and food taboos play a role in poor nutritional choices; for instance, in some cultures, pregnant women avoid iron-rich foods due to myths linking them to difficult childbirth [92]. The absence of community-based programmes further compounds these issues, particularly in rural areas where outreach efforts are minimal. Without local education initiatives to promote dietary diversification, food fortification, and supplementation, communities remain unaware of practical steps to improve nutrition [93]. Strengthening community-based awareness through public health campaigns, school nutrition education, and local interventions is essential to changing dietary behaviours and fostering long-term improvements in nutritional health.

### 5.2. Access to Nutritious Foods

Despite the importance of micronutrients, gaining consistent access to nutritious and affordable food remains a significant challenge. Several factors contribute to this persistent food insecurity and the limited availability of micronutrient-rich foods. One of the primary barriers is the high cost of nutrient-dense foods such as fruits, vegetables, dairy products, and protein sources, which are often more expensive than processed, calorie-dense alternatives. As a result, many low-income families resort to cheaper options that lack essential vitamins and minerals, ultimately compromising their nutritional status [94]. Seasonal variations and inefficiencies in food supply chains further exacerbate the problem, particularly in developing regions where access to fresh produce is inconsistent throughout the year, and poor infrastructure limits the distribution of perishable foods [95]. Rapid urbanisation has also led to a growing reliance on processed foods, which typically lack the necessary micronutrients and contribute to malnutrition [96]. Moreover, limited agricultural diversification among smallholder farmers, who often prioritise staple crops like maize and rice for economic reasons, results in the neglect of nutrient-rich crops such as legumes, leafy greens, and biofortified varieties [97]. Addressing these interrelated challenges calls for comprehensive policies that promote agricultural diversification, strengthen food distribution systems, and ensure affordable access to nutrient-dense foods for all populations.

### 5.3. Addressing Micronutrient Deficiencies: Challenges in Healthcare Accessibility and Policy Implementation

Micronutrient deficiencies are a public health concern for many, but especially for marginalised populations due to the limited availability of healthcare services and the implementation of harsh policies. Inaccessibility of healthcare on a broader basis, and especially in rural regions, often results in the unavailability of crucial microminerals and vitamins such as iron, folic acid, and Vitamin A. Furthermore, this situation is aggravated by the lack of prenatal and postnatal services provided to lactating and pregnant women [98]. Underdeveloped health systems, lack of staff, and insufficient funds suppress the success of nutrition programs in practice, and lack of funds on the part of the families further aggravates the situation in case of anaemia and stunted growth in children. Alongside this, primary healthcare professionals and other people in the health field with the unqualified title of nutritionists contribute to some of the fundamental obstacles faced by patients due to a lack of counselling about the significance of nutrition. Policy-wise, there is fragile enforcement of the guidelines provided on nutrition, such as focused food fortification, and research is needed to determine adequate funding for nutrition policies and programs, such as diet and nutrition-improving policies [99]. Silos set for identified stakeholders tend to serve more as barriers due to a lack of overlap with other silos. There is a lack of plans for what happens in a crisis to obtain food, and this further aggravates food availability in famine-like situations [100]. A fundamental reworking of these strategies is needed by improving funding for healthcare, border policies, nutrition programs, policy enforcement, and reworking the stagnant silos to allow for more collaboration between stakeholders.

#### The Vicious Cycle of Micronutrient Deficiencies and Socioeconomic Inequality

Micronutrient deficiencies are a critical aspect of hidden hunger, disproportionately affecting vulnerable populations. As illustrated in Figure 1, poverty and hunger are key drivers of limited access to nutritious food, leading to inadequate intake of essential vitamins and minerals. This deficiency manifests in a range of health issues, such as anaemia and stunted growth, as well as cognitive and physical impairments [101]. The health consequences contribute to reduced productivity and an increased economic burden, further perpetuating socioeconomic inequality [102]. In turn, socioeconomic disparities reinforce poverty and food insecurity, creating a vicious cycle that sustains micronutrient deficiencies across generations.

Addressing this issue requires targeted interventions, including food fortification, dietary diversification, and policy-driven strategies to improve food accessibility and affordability for at-risk populations. Hidden hunger, caused by insufficient intake of essential vitamins and minerals, leads to serious health issues such as anaemia, weakened immunity, and impaired cognitive development. To address these deficiencies, the table highlights strategies like dietary diversification, food fortification, and supplementation. Public health policies that enhance nutrition education and access to fortified foods are essential in mitigating hidden hunger. Micronutrient deficiencies and hidden hunger are driven by complex socioeconomic factors that hinder access to adequate nutrition. As summarised in Table 4, key determinants include poverty, conflict, climate change, urbanisation, and gender inequality. These factors collectively limit food availability, disrupt food systems, and reduce dietary diversity, disproportionately affecting vulnerable populations in regions such as Sub-Saharan Africa, South Asia, and the Middle East. Addressing these systemic challenges requires multifaceted interventions that consider both economic and social dimensions of food security.

## 6. Potential Solutions and Interventions

Addressing micronutrient deficiencies requires a multifaceted approach to tackle the root causes of inadequate nutrient intake. Effective interventions must focus on improving food quality, increasing dietary diversity, and ensuring access to essential supplements. The three key strategies to combat micronutrient deficiencies are food fortification, dietary diversification, and supplementation programmes, which, when implemented together, can significantly reduce nutrient deficiencies among vulnerable populations.

### 6.1. Food Fortification Initiatives

Food fortification is one of the most effective public health interventions for reducing micronutrient deficiencies. It involves adding essential vitamins and minerals to commonly consumed foods to improve their nutritional quality without requiring major changes in dietary habits. To mitigate the impact of micronutrient deficiencies, several intervention strategies have been implemented, each with distinct advantages and challenges (Table 5). Approaches such as food fortification, dietary diversification, supplementation, school feeding programs, and biofortification have been employed to improve micronutrient intake.

**Table 5 nutrients-17-03960-t005:** Approaches and Challenges of Dietary Diversification in Combating Micronutrient Deficiencies.

Approach	Key Benefits	Main Challenges	References
Promotion of diverse diets	Enhances intake of multiple micronutrients and supports overall health.	Limited access and affordability of diverse foods, especially in rural areas.	[103]
Home and community gardening	Improves household access to fresh, nutrient-rich foods.	Land, water, and technical constraints; seasonal variability.	[104]
Biofortified crops	Sustainable and cost-effective improvement of staple nutrient content.	Low adoption due to limited awareness and seed access.	[105]
Nutrition education	Increases nutrition knowledge and dietary behaviour change.	Cultural resistance and low literacy levels.	[106]
Agriculture–nutrition integration	Aligns agricultural production with nutrition goals	Weak coordination between the agriculture and health sectors	[107]
Addressing socioeconomic and gender barriers	Enhances equitable access to nutritious foods.	Poverty, gender inequality, and food insecurity.	[108]

While food fortification and supplementation provide immediate benefits, their success depends on effective regulation and adherence. Dietary diversification and biofortification offer more sustainable, long-term solutions but require behavioural change and policy support. School feeding programs, on the other hand, contribute to both nutritional and educational outcomes but face logistical and funding constraints. A combination of these strategies, tailored to specific regional and population needs, is crucial for effectively addressing hidden hunger and improving global nutritional health.

Essentially, food fortification falls into three types: mass fortification, targeted fortification, and biofortification. Mass fortification involves large-scale addition of micronutrients to staple foods such as wheat flour, rice, salt, sugar, and cooking oil. Examples include iodised salt, iron-fortified flour, and vitamin A-enriched cooking oil [109]. Targeted fortification focuses on foods consumed by specific groups, such as fortified infant cereals, school feeding programmes, and fortified complementary foods for pregnant women [110]. Biofortification involves breeding crops with higher nutrient content through genetic selection or biotechnology. Examples include vitamin A-enriched sweet potatoes and iron-fortified beans [111].

Food fortification offers several advantages in addressing micronutrient deficiencies. First, it is a cost-effective and scalable intervention, as it can reach large populations through existing food distribution channels without requiring extensive infrastructure or additional costs [112]. Second, food fortification requires minimal behavioural change, as fortified foods are typically incorporated into regular diets, making it easier for populations to adopt without major dietary adjustments [113]. Lastly, food fortification has demonstrated proven success in various countries. For example, nations that have implemented strong fortification policies, such as iodising salt or fortifying wheat flour with iron, have experienced significant reductions in related micronutrient deficiencies [113]. To enhance food fortification efforts, governments must collaborate with the food industry, ensure the proper enforcement of fortification policies, and conduct regular monitoring to maintain the requisite nutrient levels in fortified products.

Diversifying the diet with fruits, vegetables, legumes, nuts, dairy, and animal-source proteins without altering the existing food composition [114] is an approach that suffices as a holistic and sustainable method for resolving micronutrient deficiencies with ease. Improving local food production through home gardening, small-scale farming, and urban agriculture helps communities gain greater access to affordable and nutritious food, particularly in areas with neglected and underserved populations [115]. As described by Perera et al. [116], flexibility in post-reduction reliance on staple crops serves to ward off nutrient gaps resulting from monotonous dietary patterns. Deep-rooted issues, such as culturally restrictive diversity preferences, food scarcity, and the variability of seasonal produce, have made it very hard to realise access to fresh produce freely available all year round [117]. Therefore, long-term nutritional security is possible if all the diverse food available is made affordable by the government, in addition to agriculture-based policies, and different health institutions coordinate with local people in developing education to aid in the integration of dietary diversity.

### 6.2. Micronutrient Supplementation Programmes

Micronutrient supplementation is a targeted strategy that involves directly providing essential vitamins and minerals to individuals who are at high risk of deficiencies. These programmes are especially important for vulnerable populations such as pregnant women, infants, and malnourished individuals. Several common supplementation programmes are widely implemented across low- and middle-income countries. Vitamin A supplementation is typically administered to children aged 6 to 59 months to prevent blindness and improve immune function [118]. Iron and folic acid supplements are recommended for pregnant and lactating women to reduce the risk of anaemia and birth defects [119]. Zinc supplements are commonly administered to children to manage diarrheal diseases and boost immunity, while iodine supplements are provided to pregnant women to help prevent intellectual disabilities and thyroid disorders in newborns [120].

Micronutrient supplementation offers several benefits: it delivers rapid results by quickly raising nutrient levels, which is particularly crucial in cases of severe deficiency. The approach is also highly targeted, ensuring that vulnerable groups receive the specific nutrients they need. Additionally, supplementation can be integrated into existing healthcare services, such as maternal and child health programmes, immunisation efforts, and school health initiatives, thereby enhancing reach and efficiency [121]. However, supplementation programmes face several challenges. Accessibility remains a significant issue, particularly in rural or underserved areas where distribution systems are underdeveloped. Compliance is another concern, as individuals may fail to take supplements regularly due to limited awareness, perceived side effects, or cultural beliefs. Furthermore, although effective, supplementation is not a sustainable solution on its own. Long-term reliance on supplements must be complemented by broader efforts to improve diets and address the root causes of deficiencies [122].

To enhance the effectiveness of supplementation programmes, governments and health agencies should increase funding, strengthen supply and distribution networks, and ensure that supplementation is fully integrated into routine healthcare services. Micronutrient supplementation is a targeted approach that involves directly providing essential vitamins and minerals to individuals at high risk of deficiencies. These programmes are particularly important for vulnerable groups, including pregnant women, infants, and malnourished populations.

### 6.3. Biofortification

Biofortification is the process of enhancing the nutrient content of a crop through selective breeding, genetic engineering, or specific agronomic practices. This can be a viable and sustainable solution for populations with nutrient deficiencies that primarily rely on staple foods. Biofortified crops, i.e., vitamin A-enriched sweet potato, zinc-enriched rice, iron-enriched beans, Golden Rice (with β-carotene and ferritin) are tailored to combat vitamin A and iron deficiency in children and women. The promotion of mineral fertilisation (which increases the zinc, iron, and selenium in grains) can also help mitigate the malnutrition crisis in low- and middle-income countries (LMICs) [26,27].

### 6.4. Dietary Diversification and Education

Education, home gardening, and promoting the consumption of diverse, nutrient-rich foods can help mitigate several nutrient deficiencies in a sustainable manner. The promotion of whole foods is vital, as it helps individuals reach the recommended allowances and meet their nutrient requirements (for example, animal products are rich in iron and vitamin A; leafy greens are a good source of folate). In times of crisis, promoting immediate exclusive breastfeeding and timely complementary feeding is essential to provide small children with the essential micronutrients [6,7].

## 7. Future Directions and Research Needs

In the past few decades, formidable progress in tackling micronutrient deficiencies and hunger (MDH) has been comprehensively documented. Many reports have supported different strategies to combat MDH. In combating MDH, several countries have developed interventions, including economic growth and nutritional programs such as dietary diversification and food supplementation. In addressing fortification, biofortification has been adjudged as a prominent sustainable solution to MDH [123].

### 7.1. Food-Based Approach

Plant-based foods (PBFs) in developing countries in particular form a major part of their diets; therefore, fortification of PBFs is crucial in addressing MHG. Efforts should be intensified to biofortify agro-food crops, genetically modify plants, and employ transgenic approaches for PBFs, such as cereals, legumes/oilseeds, fruits, and vegetables [123]. Biofortification of these food crops is more relevant now that the world population is increasing, and simultaneously, stunted growth among children is at an alarming rate in the most vulnerable regions. Orange-fleshed sweet potatoes and cassava roots are among the most well-documented biofortified food crops in sub-Saharan Africa [74] and have been fully incorporated into food systems, such as bread [124]. The iron-rich beans were developed over a decade ago in Rwanda, with nearly 500,000 households incorporating them into their diets. For biofortification to be successful, several critical questions must be addressed. First, it is essential to determine whether crop breeding can effectively increase micronutrient density and significantly enhance the nutritional value of the crops. Second, the bioavailability and bio-digestibility of these added nutrients must be confirmed to ensure that they are efficiently absorbed when consumed. Third, the willingness of farmers to adopt and cultivate the newly developed crop varieties plays a vital role in the success of biofortification efforts. Ultimately, consumer acceptance of these biofortified crops is crucial to ensure widespread adoption. Addressing these questions is crucial to advancing the biofortification of staple food crops through genetic engineering, with the goal of improving micronutrient intake and combating hidden hunger [125].

### 7.2. Sustainable Food Systems

Specialised agencies of the United Nations, notably the FAO and WHO, should intensify their efforts and collaborate with the World Food Programme and the International Fund for Agricultural Development. The major aims of the collaboration should be directed at improving the general nutrient density, bioavailability, and bioaccessibility of food systems in vulnerable regions of the world (Figure 2). Moreover, by 2030, the UN-SDG2 policy on Zero Hunger aims to eradicate malnutrition and enhance human nutrition, well-being, and agricultural sustainability [125]. A comprehensive understanding of the long-term health implications of MHG and their impact on various population segments, including maternal and child health, individuals with obesity, diabetes, and cardiovascular disease [126], is necessary. Also, socio-economic factors such as food insecurity, poverty, and insufficient access to healthy or nutritious foods must be addressed. Also, increasing public awareness of the importance of a balanced diet is critical. Davies et al. [126] suggested that sufficient consumption patterns for specific foods and nutrients should be encouraged and based their assumption on the daily energy intake in the ratio 4:4:2 between breakfast, lunch, and dinner. Consumption of animal foods and fruits should be in the daytime, whereas whole grains, vegetables, dairy products, and vitamin and mineral supplementation are best consumed at dinner.

An interdisciplinary approach, involving stakeholders (such as nutritionists, food scientists, and public health experts) in the food chain, should be intensified. Formidable research should be channelled on addressing the variability among the vulnerable populations, such as children, pregnant women and those with terminal or chronic diseases that are at higher risk of hidden hunger. This is important because nutritional needs and preferences can differ significantly among vulnerable groups [126].

### 7.3. Gaps in Current Research

Despite the depth of this review, several significant areas of information remain missing and should be addressed in the near future. Notably, there is limited evidence on the impact of micronutrient health gaps on specific populations, including the effectiveness of various intervention strategies. The interaction between food production, consumption patterns, and environmental sustainability is also underexplored and deserves further investigation. In addition, the role of personalised nutrition that considers individual variability in nutritional needs and responses remains largely unaddressed. The connection between human nutrition and mental health is another area that has not been sufficiently examined, despite its growing relevance. Lastly, the issue of disparities in nutrition and health outcomes among vulnerable populations is not thoroughly reviewed. These gaps highlight the need for future research that is both comprehensive and inclusive to better inform effective nutritional interventions and policies.

### 7.4. Emerging Technologies in Nutrition

Emerging technologies refer to the continuous advancement of existing techniques that hold great potential to improve human nutrition and overall well-being. These innovations are increasingly being integrated into nutrition science and food systems. One major development is the application of artificial intelligence and machine learning algorithms to analyse large datasets to identify individual dietary requirements and food preferences for personalised nutrition planning. In addition, genomic technologies are being used to understand a person’s genetic makeup, allowing for dietary recommendations tailored to specific health conditions [126]. Another significant area of advancement is the discovery of alternative protein sources, including cultured meat, lab-grown foods, plant-based proteins, and edible insects. Technologies such as microencapsulation and nanotechnology are being utilised to develop innovative nutrient delivery systems that enhance nutrient bioavailability [128]. Furthermore, three-dimensional bioprinting technology is being explored to create customised food structures and textures, enhancing both food processing and consumer experience [125]. Additionally, the use of the Internet of Things and blockchain technology is gaining momentum in tracking food crops and products from farm to table, ensuring food safety, traceability, and reducing waste. Techniques such as high-pressure processing, cold plasma, and pulsed electric field treatment are also being employed to inactivate harmful microorganisms without compromising food quality. These emerging technologies offer promising solutions to strengthen global nutrition systems and support sustainable food security [129].

## 8. Limitations and Knowledge Gaps

There are significant limitations in the current review, which are very significant to the broader research landscape in nutritional deficiencies. Some of these limitations are geared towards the absence of comparative studies that assess the long-term economic returns and impacts on health when several of these interventions are combined. Such combinations may be fortification *plus* biofortification versus single interventions. Another important gap is the aspect of rigour in the science backing up such implementation. This involves policy research on regulatory failure and the crucial knowledge of the socio-cultural barriers to the adoption of fortified products and biofortified crops or food products by the affected communities or people. Likewise, the dependence on heterogeneous global data reveals intricacies in localising interventions and quantifying the efficacy across extremely diverse geographic, cultural and political settings.

## 9. Conclusions

Micronutrient deficiencies still pose a major global public health challenge, driven by dietary inadequacies, socio-economic disparities, and persistent gaps within food systems. Even as progress has been made in selected regions, hidden hunger remains widespread, particularly among women and children. The evidence reviewed in this manuscript further reiterates that no single intervention is adequate. Rather, meaningful and sustained improvements necessitate an integrated approach—a combination of dietary diversification, targeted supplementation, food fortification, and biofortified crops, supported by strong policies and community engagement. Improvement in local food systems will require acceleration of innovation and investment in context-specific interventions to close the remaining nutrition gaps. Prioritising these strategies can go a long way toward reducing the burden of micronutrient deficiencies and advancing global commitments to better health, resilience, and nutritional well-being.

## Figures and Tables

**Figure 1 nutrients-17-03960-f001:**
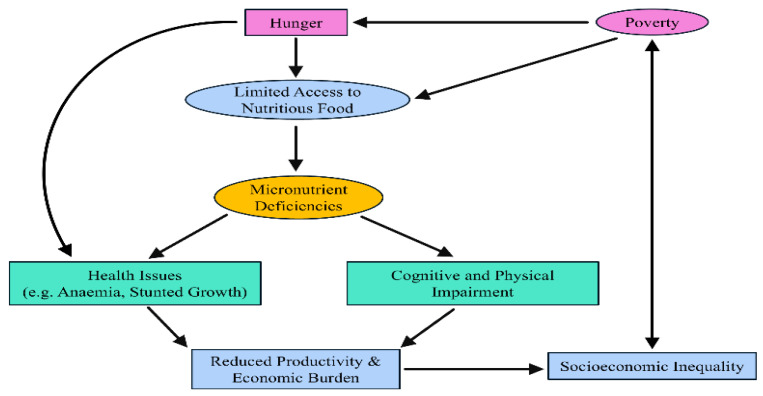
The vicious cycle of micronutrient deficiencies and economic burden in vulnerable populations.

**Figure 2 nutrients-17-03960-f002:**
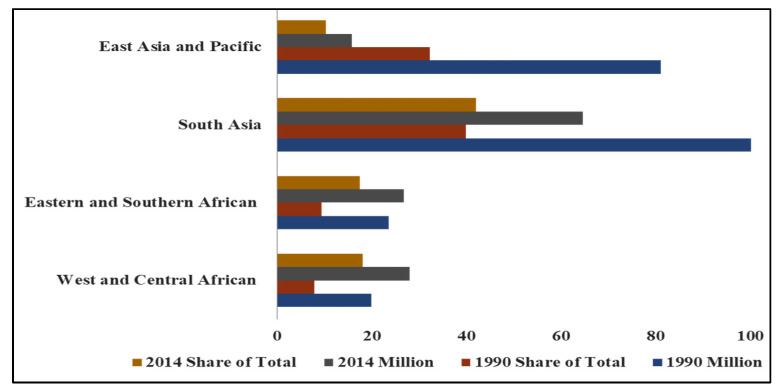
Malnutrition (Stunting)-Affected Numbers of Children and Share of Total Numbers by Regions (Adapted from [127]).

**Table 4 nutrients-17-03960-t004:** Socioeconomic factors influencing hunger and micronutrient deficiencies.

Factor	Description	Impact on Nutrition	Example Regions
Poverty	Low income affects food choices	Limited access to diverse diets	Sub-Saharan Africa, South Asia
Conflict	Displacement disrupts food systems	Inconsistent food supply	Syria, Yemen, Sudan
Climate change	Crop failures, food price spikes	Reduced availability of key nutrients	Horn of Africa, South America
Urbanization	Shift to processed foods	Lower intake of micronutrients	Emerging economies
Gender inequality	Women’s limited access to resources	Increased maternal and child malnutrition	South Asia, Middle East

## Data Availability

The original contributions presented in the study are included in the article, further inquiries can be directed to the corresponding author.

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
