# Peer review of "Beyond Calories: Addressing Micronutrient Deficiencies in the World’s Most Vulnerable Communities—A Review"

_nutrients, 2025, doi:10.3390/nu17243960_

Round 1

Reviewer 1 Report

Comments and Suggestions for Authors

This manuscript describes a very important issue, not only for the world’s most vulnerable communities, but also for affluent countries.

This paper might be discussed:

Food in the Anthropocene: the EAT–Lancet Commission on healthy diets from sustainable food systems W Willett, J Rockström, B Loken, M Springmann… - The lancet, 2019 - thelancet.com … quality diets that cause micronutrient deficiencies and contribute to a substantial rise in the
incidence of diet-related obesity and diet-… Unhealthy diets pose a greater risk to morbidity and

This article has interesting results regarding 38 types of food in 8 dietary patterns and their associations with risk of chronic diseases. The main finding was the beneficial effect of diets that reduce inflammation and insulin resistance. As diabetes is an important global issue, these findings should be of interest.

Optimal dietary patterns for prevention of chronic disease.

Wang P, Song M, Eliassen AH, Wang M, Fung TT, Clinton SK, Rimm EB, Hu FB, Willett WC, Tabung FK, Giovannucci EL. Nat Med. 2023 Mar;29(3):719-728. doi: 10.1038/s41591-023-02235-5.

I suggest that magnesium might be added to the list of micronutrients to discuss. Google Scholar can be searched to show which papers have the highest citation numbers.

Iodine is also important, especially for women for good pregnancy outcomes, thyroid function, and prevention of breast cancer. It might be mentioned as well. Iodized salt is often used as a source.

Another factor to consider is the reduction in important trace minerals in soils used for agriculture as well as erosion of agricultural soil.

Meat eggs are also sources of vitamin D. For meat, it is in the form of 25-hydroxyvitamin D. However, not sure that meat should be included in the recommendations.

Regarding calciferol, Visser et al., 2006 is rather old. Papers published within the past five years are generally preferable.

The list of the beneficial effects of vitamin D includes many more health outcomes than indicated in de Martinis et al., 2021. See, e.g.,

Vitamin D: Evidence-Based Health Benefits and Recommendations for Population Guidelines.

Grant WB, Wimalawansa SJ, Pludowski P, Cheng RZ. Nutrients. 2025 Jan 14;17(2):277. doi: 10.3390/nu17020277.

The global vitamin D deficiency rate is closer to 50% than 14%. Also, please define the 25(OH)D concentration for deficiency (50 nmol/L)?

See, e.g.

A systematic review and meta-analysis of circulating 25-hydroxyvitamin D concentration and vitamin D status worldwide E Dunlop, NM Pham, D Van Hoang… - Journal of Public …, 2025 - academic.oup.com

Worldwide vitamin D status N van Schoor, R de Jongh, P Lips - Feldman and Pike's Vitamin D, 2024 - Elsevier

D > 50 nmol/L) in less than 50% of the world population at least in winter. Prevention of vitamin
D deficiency … , fortification of foods with vitamin D, and the use of vitamin D supplements. …

Re Tocopherol. There are four varieties of tocopherol: alpha, beta, gamma, and delta. Many if not most of the clinical trials used alpha-tocopherol and most of them did not find good results. The reason is that there are receptors for all four varieties of tocopherol and if alpha binds to receptors looking for the other three, there is no beneficial effect. Mixed tocopherol supplements are more expensive than alpba-tocopherol. Thus, natural vitamin E sources such as nuts might be proposed.

Not discussed much in the manuscript are two important topics: obesity and ultraprocessed foods. I suggest considering adding them.

Regarding obesity, this is an important overview:

Obesogens: a unifying theory for the global rise in obesity.

Heindel JJ, Lustig RH, Howard S, Corkey BE. Int J Obes (Lond). 2024 Apr;48(4):449-460. doi: 10.1038/s41366-024-01460-3.

It states that the energy-balance hypothesis does not seem to apply to developed countries now; however, it does apply to countries that are increasing their caloric intake or supply.

Line 592: micronutrient deficiencies and

hunger (MHG)

Comment: what does G stand for?

Figure 2. Suggest that the box with “Health issues” use “Physical Health Issues”

Author Response

Dear Reviewer.

We appreciate your efforts in improving the standard of this manuscript. We have attended to all the comments, and we have attached our responses. 

Reviewer 2 Report

Comments and Suggestions for Authors

All the comments were included in the attached file.

Comments on the Quality of English Language

The English is on a very high level, and only a few minor issues were detected.

Author Response

Dear Reviewer,

We thank you for your constructive comments to make the manuscript better. Thank you very much. We have provided responses to the comments, and we attached the response. We hope our responses addressed all the comments.

Kind regards,

Peter (on behalf of all the authors)

Reviewer 3 Report

Comments and Suggestions for Authors

I have several suggestions.

  1. I do not find Figure 1 useful and in my view it can be deleted.
  2. The tables in general need to be improved - many of the words are hyphenated which makes it difficult to read - this is especially true of Tables 1, 3 and 5.

Author Response

Dear Reviewer,

We appreciate all the comments given to improve the standard of our manuscript. We have provided responses to the comments and made corrections accordingly on the manuscript. We hope our responses addressed all the comments. Thank you very much.

Kind regards,

Peter (on behalf of all the authors)

Reviewer 4 Report

Comments and Suggestions for Authors

The paper aims to offer perspectives on addressing hidden hunger and enhancing global health. It primarily focuses on vitamin A and D deficiencies and four minerals (Fe, I, Ca, Zn). The review could make a significant contribution to the field by presenting examples of best practices to support socially and economically vulnerable groups. It includes relevant information on potential mineral and vitamin deficiencies and their related health issues, along with possible solutions. The data mainly relate to food (bio)fortification.

To improve the article's scientific rigor and alignment with journal standards, the authors might consider the following suggestions:

  • Revise the citation format;
  • Add discussion of study limitations;

Besides the suggested possible interventions, they could also explore resilient food systems, such as smart agriculture (encouraging crop diversification and family and community gardens), improvements to the value chain, and measures to reduce food losses.

Author Response

Dear Reviewer,

We thank you for painstakingly making constructive comments to improve the standard of the manuscript. We have provided responses to the comments and also made the necessary corrections to the manuscript. Attached are the responses to the comments. We hope our responses addressed all the comments.

Kind regards,

Peter (on behalf of all the authors)

Round 2

Reviewer 2 Report

Comments and Suggestions for Authors

Thank you for your resubmission. You have addressed many of the previous comments, but the article still contains numerous repetitions. The text, instead of being shorter, is longer after revision. On the other hand, the interesting part, on the effects of the intervention, was not expanded. Besides addressing former comments, a minor change to add are more contrasting colours in Figure 3. 

Comments on the Quality of English Language

The English is on a very high level, and the text reads well besides repetitions that could be removed.

Author Response

Dear Reviewer,

We appreciate your effort in improving the standard of this review manuscript. We have provided a step-by-step response to the comments. We believe the comments had helped use to improve the standard of the manuscript. 

Reviewer comments and Response

Thank you for your resubmission. You have addressed many of the previous comments, but the article still contains numerous repetitions. The text, instead of being shorter, is longer after revision. On the other hand, the interesting part, on the effects of the intervention, was not expanded. Besides addressing former comments, a minor change to add more contrasting colours in Figure 3. 

Response:

We thank the reviewer for these comments. We have added another short paragraphs to the intervention part of the manuscript. In the case of repetitions, we found that many of the points mentioned are most common among high-risk groups, which led to the repetitions you observed. Figure 3 was adapted from an article for which we did not have the raw data to replot the graph. We have removed some parts of the graph which have little or no relevance to the discussion.

Thank you.

Kind regards

Peter (For all the authors).
